# Highly Efficient and Stable Self-Powered Perovskite Photodiode by Cathode-Side Interfacial Passivation with Poly(Methyl Methacrylate)

**DOI:** 10.3390/nano13030619

**Published:** 2023-02-03

**Authors:** Wonsun Kim, JaeWoo Park, Yushika Aggarwal, Shital Sharma, Eun Ha Choi, Byoungchoo Park

**Affiliations:** 1Department of Electrical and Biological Physics, Kwangwoon University, Seoul 01897, Republic of Korea; 2Department of Plasma-Bio Display, Kwangwoon University, Seoul 01897, Republic of Korea

**Keywords:** photodetector, polymeric passivation layer, organic/inorganic hybrid perovskite

## Abstract

For several years now, organic–inorganic hybrid perovskite materials have shown remarkable progress in the field of opto-electronic devices. Herein, we introduce a cathode-side passivation layer of poly(methyl methacrylate) (PMMA) for a highly efficient and stable self-powered CH_3_NH_3_PbI_3_ perovskite-based photodiode. For effective noise–current suppression, the PMMA passivation layer was employed between a light-absorbing layer of CH_3_NH_3_PbI_3_ (MAPbI_3_) perovskite and an electron transport layer of [6,6]-phenyl-C61-butyric acid methyl ester. Due to its passivation effect on defects in perovskite film, the PMMA passivation layer can effectively suppress interface recombination and reduce the leakage/noise current. Without external bias, the MAPbI_3_ photodiode with the PMMA layer demonstrated a significantly high specific detectivity value (~1.07 × 10^12^ Jones) compared to that of a conventional MAPbI_3_ photodiode without a PMMA layer. Along with the enhanced specific detectivity, a wide linear dynamic response (~127 dB) with rapid rise (~50 μs) and decay (~17 μs) response times was obtained. Furthermore, highly durable dynamic responses of the PMMA-passivated MAPbI_3_ photodiode were observed even after a long storage time of 500 h. The results achieved with the cathode-side PMMA-passivated perovskite photodiodes represent a new means by which to realize highly sensitive and stable self-powered photodiodes for use in developing novel opto-electronic devices.

## 1. Introduction

Energy-harvesting photovoltaic (PV) technologies based on hybrid organic/inorganic perovskite materials have attracted the interest of many research groups due to the excellent optoelectronic characteristics, high charge carrier mobility, tunable bandgaps, and high diffusion ranges of these materials. Moreover, they are simple, operate at low temperatures, are solution processability, and have a low fabrication cost [1,2,3,4,5,6]. Given these characteristics, hybrid organic/inorganic perovskite materials are utilized in various optoelectronic devices such as PV or solar cells, light-emitting diodes, and especially in photodetectors [7,8,9]. In recent years, CH_3_NH_3_PbI_3_ (MAPbI_3_) organohalide perovskite photodetectors have shown significantly increased photo-detecting capabilities with rapid response times, with detectivity (*D**) values of ~7.8 × 10^12^ Jones (Jones = cm Hz^1/2^/W) and responsivity (*R*) values close to 470 mA/W [10,11,12,13]. These results are slightly lower than or comparable to those of conventional silicon photodetectors (*D** ~1 × 10^13^ Jones) [14].

Despite the advent of these new types of device structures and novel materials for improving the performance capabilities of perovskite-based devices, deeply rooted problems including defect-rich and distorted lattice structures in the bulk and surfaces of the MAPbI_3_ perovskite layers remain unsolved [15,16,17,18,19,20]. Specifically, defects in perovskite layers hamper the charge transport and extraction processes, causing interfacial recombination losses between the light-absorbing layer and adjacent charge transport layers and/or electrodes [17,18,19,20]. Due to these losses, much work remains for further improvements of the optoelectronic device performance and stability over conventional silicon-based devices [21,22].

Among the attempts to improve the interface quality of perovskite layers in PV cells, passivation layers consisting of functional organics or polymers such as poly(ethylene oxide) and poly(methyl methacrylate) (PMMA) have been suggested and successfully demonstrated. These were found to possess effective interface engineering based on their chemical and physical/electrical advantages [17,18,19,23,24]. Recently, the present authors also showed a self-powered MAPbI_3_ perovskite-based photodiode device with an anode-side passivation layer of PMMA at the interface between a MAPbI_3_ light-absorbing layer and a hole-transport layer (HTL) [25]. The anode-side PMMA passivation layer could reduce carrier recombination losses and noise currents. Thus, high specific detectivity (~0.81 × 10^12^ Jones) was achieved even without external bias [25].

Nonetheless, the introduction of a PMMA passivation layer between the light-absorbing layer and the electron-transport layer (ETL) for perovskite photodiodes has not been fully investigated and remains poorly understood [9,20]. Unlike the formation of the aforementioned anode-side passivation layer [25], a cathode-side passivation layer can be simultaneously formed during the antisolvent process used to crystallize the MAPbI_3_ layer [26]. Even without an additional coating process, issues related to the grain boundary and the problem of defects at the interface between the perovskite light-absorbing layer and ETL can be solved to realize highly efficient perovskite photodiodes. Additionally, dark and/or noise currents, which directly affect the responsivity and specific detectivity of photodetectors, can also be controlled by the cathode-side passivation layer. Thus, it is crucial to select proper interface engineering techniques for the perovskite light-absorbing layer, as doing so can improve the photocurrent and reduce the leakage and/or noise currents.

Herein, a high-performance self-powered MAPbI_3_ perovskite photodiode with a cathode-side passivation layer of PMMA was systematically investigated. To assess the optoelectronic characteristics, we studied the responsivity, detectivity, and linear dynamic response range of MAPbI_3_ perovskite photodiodes under a self-powered condition. Along with these observations, we measured the response times (rise/decay times) and stability characteristics of the photodiodes. Furthermore, we evaluated the device characteristics of the reference perovskite photodiodes without a PMMA passivation layer to support our results. The obtained results highlight the superior device performance of the perovskite photodetector with cathode-side PMMA passivation compared to a conventional reference device as well as previous perovskite photodetectors that rely on anode-side PMMA passivation.

## 2. Materials and Methods

### 2.1. Materials

Nickel(II) nitrate hexahydrate (Ni(NO_3_)_2_·6H_2_O, 99.999%), anhydrous dimethyl sulfoxide (CH_3_)_2_SO, DMSO, 99.9%), *N*,*N*-dimethylformamide (HCON(CH_3_)_2_, DMF, 99.8%), anhydrous mono chlorobenzene (C_6_H_5_Cl, CB, 99.8%), and anhydrous isopropyl alcohol ((CH_3_)_2_CHOH, IPA, 99.7%) were purchased from Sigma-Aldrich Korea (Seoul, Republic of Korea) PMMA ((C_5_O_2_H_8_)_n_, 950,000 M_w_) was purchased from Kayaku Advanced Materials (Westborough, MA, USA). Methyl ammonium iodide (CH_3_NH_3_I, MAI) and lead(II) iodide (PbI_2_, 99.9985%) for the perovskite precursor were purchased from Greatcell Solar (Queanbeyan, NSW, Australia) and Alfa Aesar (Haverhill, MA, USA), respectively. Ethylene glycol (HO(CH_2_)_2_OH, EG, 99%) and IPA for the nickel oxide (NiO_x_) precursor were purchased from Daejung Chemicals and Metals (Siheung, Republic of Korea). Phenyl-C61-buthyric acid methyl ester (C_72_H_14_O_2_, PCBM_60_) was purchased from Nano-C (Westwood, MA, USA). Bathocuproine (C_26_H_20_N_2_, BCP, 98%) and a colloidal suspension of ZnO nanoparticles were procured from Tokyo Chemical Industry Co. Ltd. (Tokyo, Japan) and Nanograde (Zurich, Switzerland), respectively. All materials were used as received without further purification.

### 2.2. Methods

Pre-patterned 80-nm-thick indium tin oxide (ITO, 20 Ω/sq) layers on glass substrates were used as transparent anodes for the photodiodes. The ITO substrates used here were ultrasonically cleaned with ethanol, a detergent, and deionized (DI) water, after which they underwent an ultraviolet ozone treatment in an oven for 10 min.

For the NiO_x_ HTL, a precursor solution with 29 mg of Ni(NO_3_)_2_·6H_2_O dissolved in mixed solvents of EG and IPA at a 6:4 volume ratio was prepared by stirring for 3 h. The precursor solution was then spin-coated onto the ITO substrate at 2600 rpm for 40 s, followed by 60 min of annealing at 300 °C to form the a 10-nm-thick NiO_x_ HTL. Subsequently, the substrates coated with the NiO_x_ layer were placed in a nitrogen-filled glovebox.

To fabricate the perovskite light-absorbing active layers of MAPbI_3_, an antisolvent-assisted spin-coating method was used [25,26]. The perovskite precursor solution was prepared by dissolving PbI_2_ and MAI at a 1:1 molar ratio in a mixed polar solvent of DMF and DMSO at an 8:2 volume ratio, with the precursor solution then stirred overnight. In the nitrogen-filled glovebox, the precursor solution was spin-coated onto the NiO_x_-coated substrate at 4400 rpm for 30 s. During the spinning process, anhydrous CB or CB containing 0.05 wt% of PMMA was dropped onto the perovskite precursor-coated substrate as an antisolvent at a dripping delay time of 5–8 s after the spinning process had started. When the dripping time was changed from 5–8 s, the film quality of the fabricated perovskite layer was degraded and the film became inhomogeneous and hazy. Hence, in this study, we set the dripping time to 5–8 s to fabricate a uniform and homogeneous perovskite precursor layer without or with a cathode-side PMMA passivation layer. The perovskite precursor layer was then dried at room temperature for 5 min and subsequently annealed at 100 °C for 20 min to crystallize the MAPbI_3_. The thicknesses of the MAPbI_3_ layers studied here were nearly identical to each other (~250 nm), and the thickness of the PMMA layer was estimated to be less than 5 nm.

Thereafter, PCBM_60_ in CB and a colloidal suspension of ZnO nanoparticles were subsequently spin-coated above the coated layers, resulting in a uniform 50-nm-thick PCBM_60_ layer and a 20-nm-thick layer of ZnO as ETLs. The substrates were then transferred to a vacuum chamber for thermal evaporation of the 12-nm-thick BCP layer and 70-nm-thick Al cathode layer, subsequently at a base pressure below 2.0 × 10^−6^ torr. Thus, a photodiode with the structure of [ITO/NiO_x_/MAPbI_3_/PMMA/PCBM_60_/ZnO/BCP/Al] was fabricated. All fabricated photodiodes had an active area of 6 mm^2^.

### 2.3. Characterization

The surface morphology of the MAPbI_3_ perovskite layer was analyzed with a scanning electron microscope (Inspect F50, FEI Company, Hillsboro, OR, USA). ImageJ software (National Institutes of Health, Bethesda, MD, USA) was used to analyze the grain size distributions in the MAPbI_3_ perovskite layers from the observed SEM images.

The UV–Visible optical adsorption spectra were studied using an UV–Vis spectrometer (Agilent 8453 Diode Array UV-VIS, Agilent Technologies, Inc., Santa Clara, CA, USA). The contact angles of the fabricated MAPbI_3_ films were measured with a contact angle goniometer (Ossila, London, UK).

A monochromatic light source of a 637 nm diode laser (model COMPACT-100G-637-A, 100 mW, maximum internal modulation frequency: 50 kHz, World Star Tech, Markham, OT, Canada) was used to assess the performance of the photodetector. Keithley source meters (models 2400 and 2636, Tektronix, Inc. Beaverton, OR, USA) were used to assess the current-versus-voltage (*J*–*V*) characteristics. An incident photon-to-current conversion efficiency (IPCE) measurement system (ORIEL IQE-200, Newport, Irvine, CA, USA) was used to measure the spectral responsivity, R(λ), of the photodiode.

The noise current levels of the photodiode were obtained from the fast Fourier transform of the measured dark currents of the photodiode as a function of time using the 2636 Keithley source meter operated at a sampling rate of 1 kHz. The 3 dB cutoff bandwidths of the photodiodes were estimated from the logarithmic transform of their normalized photo responses, measured as a function of the modulation frequency of the irradiated light using the 637 nm laser system.

## 3. Results

### 3.1. Characteristics of MAPbI_3_ Layers with a PMMA Passivation Layer

A schematic illustration of the passivation layer of the PMMA on a MAPbI_3_ perovskite layer is shown in Figure 1a. As illustrated in the figure, PMMA dissolved in an antisolvent can simply be spin-coated during the formation of a 250-nm-thick MAPbI_3_ perovskite light-absorbing layer. To verify the formation of the PMMA passivation layer on the perovskite layer, first, we observed the contact angles of DI water on the fabricated films (Figure 1b). As shown in the figure, the water contact angle on the NiO_x_/MAPbI_3_/PMMA perovskite film (sample) was 37.4°, significantly higher than that (~26.8°) of the NiO_x_/MAPbI_3_ perovskite film (reference). This increase in the water contact angle mainly stems from the intrinsic hydrophobicity of the PMMA layer in the sample film, providing evidence of the simple and reliable fabrication method of the PMMA passivation layer on the perovskite layer. Thus, such a hydrophobic PMMA passivation layer may affect the growth of the grains of the underlying MAPbI_3_ perovskite by tailoring the nucleation of the perovskite crystal growth [3,13,27]. It is noteworthy that the coated PMMA polymers mainly existed at the perovskite/ETL interface, with some of the polymer possibly present inside the perovskite layer, similar to an earlier conjugated polymer that formed an interfacial passivation layer [20].

To assess the effect of the PMMA passivation layer on the growth of grains for the MAPbI_3_ perovskite films, scanning electron microscopy (SEM) was utilized. The surface morphologies obtained from the SEM observations are shown in Figure 1c. As shown in these SEM images, the surfaces of both perovskite films were fairly smooth and showed well-packaged grains. However, compared to the reference film without a passivation layer, it is clear that the introduction of the PMMA layer affected the increment of the grain size while also decreasing the number of grain sites in the sample perovskite film.

For an additional quantitative analysis, ImageJ software was used to adjust the contrast of the SEM images of the MAPbI_3_ perovskite layers without/with a PMMA passivation layer (Figure 1c) and to identify the borders of the grains in the SEM images. Then, utilizing the black and white threshold of each adjusted SEM image, it was possible to estimate the range of the domain sizes of the surrounding grains. Histograms of the domain size distributions for MAPbI_3_ perovskite layers without/with a PMMA passivation layer are presented in Figure 1d. As shown in this figure, the average grain size of the reference film was ~151 nm, while that of the sample film was ~196 nm, clearly demonstrating that the average grain size of the MAPbI_3_ layer was increased significantly due to the presence of the PMMA passivation layer. From these SEM studies, it was confirmed that the generation of unnecessary MAPbI_3_ nucleation could be significantly suppressed by the coordinate bonding of the carbonyl of the PMMA polymer with the uncoordinated Pb ions of MAPbI_3_ [28,29,30]. Thus, the PMMA passivation layer enables an increase in the MAPbI_3_ grain size while also mitigating film defects such as spikes and/or pinholes [31,32].

Apart from the increased grain size of the MAPbI_3_ perovskite layer, the PMMA passivation layer can effectively cover defects and grain boundaries in the perovskite layer, possibly providing an effective passivation route for interface defects between the MAPbI_3_ layer and the adjacent ETL [33,34]. Moreover, due to the improved interface quality of the MAPbI_3_ layer stemming from the PMMA, the reduction in the recombination loss of charge carriers can improve the charge carrier extraction from the MAPbI_3_ layer to the charge transport layers [31,32,34,35,36]. Thus, given these properties of the MAPbI_3_ layer in the sample films, we hold that the introduction of a PMMA passivation layer can effectively improve the film and interface qualities of the MAPbI_3_ layer, likely leading to an improvement in the optoelectronic performance of perovskite devices, even without any critical changes in the optical characteristics such as the optical absorption capabilities (Figure 1e) [25]. To confirm this, the optical absorption characteristics of the reference and sample perovskite films were also investigated, as shown in Figure 1e. In this figure, strong absorption in the visible range of 450~700 nm for the reference and sample films could be observed. Notably, the sample film showed an optical absorption nearly identical to that of the reference perovskite film, proving that the PMMA passivation layer did not significantly change or deteriorate the optical absorption properties of the perovskite layer. Thus, by introducing the PMMA passivation layer onto the perovskite layer, modification of the interface between the MAPbI_3_ layer and the ETL can be anticipated.

### 3.2. Characterization of Perovskite Photodiodes without and with a PMMA Passivation Layer

Inspired by the desirable passivation effect of PMMA described above, we fabricated MAPbI_3_ photodiodes with the device structure illustrated in Figure 2a. As shown in the figure, ITO acts as a transparent anode, with NiO_x_ used as a HTL, MAPbI_3_ as a perovskite light-absorbing layer, and PMMA as the cathode-side passivation layer. PCBM_60_, ZnO nanoparticles and BCP were utilized as the ETLs and Al was used as the top cathode electrode.

To evaluate the device performance of the fabricated MAPbI_3_ photodiodes without (reference) and with a PMMA passivation layer (sample), the dark-current densities (*J*_dark_) were investigated as a function of the applied voltage (*J*_dark_-*V*). As indicated in Figure 2b, excellent diode characteristics with large rectifying ratios (*RR*s) and low leakage current densities were found in these photodiodes. Due to the suppression of the leakage currents by the PMMA passivation layer, a somewhat higher *RR* value, 6.1 × 10^5^ (at 1.0 V), for the sample was obtained compared to that (5.3 × 10^5^) of the reference, indicative of the useful film properties of the MAPbI_3_ perovskite films, as above-mentioned. It is also noteworthy that the observed *RR* values were one order higher than the corresponding values in the literature [13]. Similarly, the PMMA passivation layer in the sample led to a reduced average *J*_dark_ value of 5.1 × 10^−7^ mA/cm^2^ at zero applied voltage, lower than that (7.3 × 10^−7^ mA/cm^2^) of the reference without a PMMA layer, providing evidence of the suppressed leakage currents caused by the improved interfacial quality of the perovskite light-absorbing layer.

Furthermore, for an in-depth examination of the perovskite photodiodes presented here, their trap-filled limit voltages (*V*_TFL_s) were estimated from the *J*_dark_–*V* curves on a log–log scale (Figure 2c) [13,20,32,37,38]. The obtained *V*_TFL_ values for the reference and sample were ~0.49 and 0.47 V, respectively. Based on the *V*_TFL_ values, the trap density states (*n*_trap_) can also be estimated using the space–charge–limited current model, as follows [38,39]:(1)$$V_{\mathrm{TFL}}=\frac{qn_{\mathrm{trap}}L^{2}}{2\varepsilon_{0}\varepsilon_{p}}\;$$

In this equation, *q* is the elementary charge, *L* is the thickness of the perovskite layer, while *ε*_0_ and *ε*_p_ are the dielectric constants of the vacuum and perovskite, respectively. The estimated *n*_trap_ values were around 1.30 × 10^15^ cm^−3^ for the reference and 1.25 × 10^15^ cm^−3^ for the sample, indicating that the PMMA layer in the sample could effectively passivate defects at the interface between the perovskite layer and the ETL.

Subsequently, we measured the photocurrents as a function of the bias voltage in the range of ±1.2 V (*I*_light_ -*V*) for several input power levels (*P*s) of incident laser light with a wavelength of *λ* = 637 nm, as shown in Figure 3a. In the figure, the open-circuit potential (*V*_OC_) and the short-circuit current (*I*_SC_) increased considerably as the input power of the incident light was increased. For a detailed comparison, the built-in potentials (*V*_bi_s) of the reference and sample were obtained using the Shockley diode model equation, expressed as follows,
(2)$$V_{\mathrm{bi}}=-\left(\frac{nk_{B}T}{e}\right)\ln J_{0},\;$$
where *k*_B_ is the Boltzmann’s constant; *T* is the temperature; *n* is the ideality factor; and *J*_0_ is the reverse-saturated current density [13,40,41,42]. The estimated values of *V*_bi_ and *n* were approximately 0.58 V and 1.49, respectively, for the sample, slightly higher than the *V*_bi_ value of approximately 0.57 V and the *n* value of about 1.47 for the reference. As the thickness of the PMMA layer was considered to be less than ~5 nm, it is clear that the PMMA passivation layer did not degrade the built-in potential despite its intrinsic insulating properties. It should be noted that the short-circuit current *I*_SC_ and built-in potential *V*_bi_ began to decrease noticeably as the thickness of each functional layer, especially the PMMA layer or the MAPbI_3_ layer, was changed from its optimized thickness.

Next, the *I*_SC_ values of the reference and sample were determined using Figure 3a and are plotted as a function of the input power of the incident laser light (Figure 3b). It is clear from the figure that even under the self-powered condition (at zero bias voltage), large numbers of charge carriers become separated by the built-in potential, resulting in a considerable increment of *I*_SC_. With the power law of
(3)$$I_{\mathrm{SC}}=\kappa \times P^{\theta },$$
where *κ* is a proportional constant and *θ* is the power law index, the trap states existing in the MAPbI_3_ perovskite light-absorbing layer studied here can be analyzed [13,20,43]. We obtained *θ* values for both the reference and sample using the power law above with the best fitted parameter values, determining an *θ* value of 0.999 for the sample, which is much closer to *θ* = 1.0 for an ideal photodiode compared to the value of *θ* = 0.984 for the reference without a PMMA layer. Hence, the cathode-side PMMA passivation layer can effectively decrease the number of trap states and reduce the second-order recombination loss under a short-circuit condition (i.e., self-powered condition).

Next, based on the photocurrent data in Figure 3b, the responsivity at the wavelength of *λ* (*R*_λ_) for the reference and sample were estimated using the equation.
(4)$$R_{\lambda }=\left(\frac{I_{\mathrm{PH}}}{P}\right)\;\mathrm{and}\;I_{\mathrm{PH}}=I_{\mathrm{light}}-I_{\mathrm{dark}}$$
where *I*_PH_ denotes the net photocurrent [7,9,12,13,14,20]. The responsivity *R*_637_ value at the wavelength *λ* = 637 nm of incident light under zero bias voltage as obtained here was approximated as 360 mA/W for the sample with the PMMA passivation layer, higher than that (~352 mA/W) of the reference at zero bias voltage.

To evaluate the photodiode performance over a wide light intensity range, we also calculated the linear dynamic range (*LDR*) of the photodiodes using the equation [7,9,12,14,20]:(5)$$LDR=20\log\left(\frac{I_{\mathrm{PH}}}{I_{\mathrm{dark}}}\right)$$

Even at zero bias voltage, a notably higher *LDR* value was realized, ~127 dB, for the sample compared to that (~124 dB) of the reference. These high values of *R*_λ_ and *LDR* are clear evidence of the excellent photoelectric conversion ability and remarkably good linearity over a wide range of incident light intensity for the sample with the cathode-side PMMA passivation layer. It should also be noted that the sample device showed a relatively high *LDR* value compared to those of previous perovskite thin-film photodetectors and industrial silicon photodetectors (90~120 dB). Given the nearly ideal value of *θ* and the high values of *R*_λ_ and *LDR*, the cathode-side PMMA passivation layer can serve to realize trap-less perovskite photodiodes.

For a further evaluation of the photodiode performance, the *R*_λ_ spectra were also measured using an IPCE system in the self-powered condition, as presented in Figure 4a. This figure indicates that the peak *R*_λ_ value of ~401 mA/W for the sample was notably higher than that (386 mA/W) for the reference and higher than those of the MAPbI_3_-based photodiodes in the literature [9,13,25]. This finding clearly shows that the improved interface of the MAPbI_3_ layer for efficient charge extraction is crucial when attempting to realize high responsivity from perovskite photodiodes.

Subsequently, we analyzed the noise current (*i*_n_) of the photodiodes. At an ambient temperature, the dark currents (*I*_dark_) were measured over time with no voltage applied. Subsequently, the dark currents were fast Fourier transformed and plotted as a function of the frequency, as shown in Figure 4b [7,10,11,20,44], where the photodiodes exhibited white noise unrelated to the frequency in the ranges observed here. Additionally, as shown in the figure, the noise level of each photodiode was evaluated at a bandwidth of 1 Hz. The estimated noise level of the sample was ~0.09 pA Hz^−1/2^ at a bandwidth of 1 Hz, somewhat lower than that (0.96 pA Hz^−1/2^) of the reference. This indicates that the proposed cathode-side PMMA passivation layer clearly suppresses the noise level by passivating unwanted film defects at the interface between the MAPbI_3_ layer and the PCBM_60_ ETL.

Next, the obtained values of *i*_n_ and *R*_λ_ were applied to assess the noise equivalent power (*NEP*) in an effort to determine the minimum detectable incident optical power using the relationship of [7,9,11,12,14,45,46]:(6)$$NEP=\frac{i_{n}}{R_{\lambda }}\;$$

The obtained minimum *NEP* value for the sample was ~230 fW/Hz^1/2^, notably less than that (~2520 fW/Hz^1/2^) for the reference at zero bias voltage. This clearly indicates that the cathode-side PMMA passivation layer can enhance the low-power-detection ability of the MAPbI_3_ photodiode compared to a non-passivated MAPbI_3_ photodiode.

Based on the *NEP* values presented above, the specific detectivity *D*^*^ values of the photodiodes were also evaluated under the self-powered condition in order to evaluate their weak-signal detection capacities. This was carried out with the following equation,
(7)$$D^{*}=\frac{\sqrt{A\cdot B}}{NEP},$$
where A and B are the active area of the device and the 1 Hz specific bandwidth, respectively [7,9,10,11,12,13,14,20,44,45,46,47,48,49]. In the self-powered condition, the estimated specific detectivity *D*^*^ spectra for the reference and sample using Equation (7) are plotted in Figure 4c. As shown in this figure, while the peak value of *D*^*^ was ~0.97 × 10^11^ Jones for the reference, the peak value of *D*^*^ for the sample was ~1.07 × 10^12^ Jones, which is more than eleven times greater for weak-signal detection compared to the reference. Hence, the suppression of unnecessary leakage/noise currents by the cathode-side PMMA passivation layer clearly increased the photodetector performance outcomes. Additionally, the *D*^*^ value of 1.07 × 10^12^ Jones for the sample was conspicuously higher than those (*D*^*^ value ~0.81 × 10^12^ Jones) of the MAPbI_3_ photodiodes in the literature with an anode-side PMMA passivation layer at the interface between the MAPbI_3_ perovskite light-absorbing layer and the NiO_x_ HTL; this was also comparable to those of commercial silicon photodetectors [14,20,25,47].

As another comparison, we also estimated the conventional simplified specific detectivity *D*^*^ values of the reference and sample based on the shot noise ($i_{n,s}$), as estimated from the dark-current density ($J_{\mathrm{dark}}$) with the simple relationship of $D^{*}\;\sim \;R_{\lambda }/\sqrt{2eJ_{\mathrm{dark}}}$ [7,9,13,14,48,49]. The estimated peak value of the simplified *D*^*^ for the reference was ~2.5 × 10^13^ Jones. In contrast, the peak value of simplified *D*^*^ for the sample showed an increase to ~4.5 × 10^13^ Jones. It should also be noted that the simplified *D*^*^ value for the sample was significantly higher compared to earlier values for other MAPbI_3_ thin-film-based photodiodes previously reported in the literature [25,48,49]. However, such simplified *D*^*^ values may be overestimated because the shot noise levels were lower than their measured noise current values, as indicated by the dotted lines in Figure 4b [49].

To assess the dynamic characteristics of the photodiodes studied here, the temporal responses were investigated at zero bias voltage by irradiating monochromatic light (*λ* = 637 nm, *P* = 190 μW) modulated at a frequency of 2 kHz (Figure 5a). In Figure 5a, the measured temporal responses of the sample clearly indicated higher photocurrents than those of the reference, as expected. The response times of the rise (*τ*_r_) and decay (*τ*_d_) times were determined as the time intervals required to change the photocurrent signal amplitudes between 10% (*I*_10_) and 90% (*I*_90_) up the rising and decay edges of the signal curves, respectively; *τ*_r_ and *τ*_d_ for the sample were ~50 and ~17 μs, respectively, similar to those (~49 and ~18 μs) of the reference. Moreover, the obtained response times for the sample and reference were considerably shorter than those of MAPbI_3_-based photodiodes in the literature [13].

Next, the 3 dB cutoff bandwidths (*f*_-3dB_) for the photodiodes were measured. These values were *f*_-3dB_ = ~1.40 × 10^4^ and ~1.33 × 10^4^ Hz for the reference and sample, respectively (Figure 5b). These dynamic photo responses of the photodiodes indicate that the limitation of the response bandwidths in the devices may stem mainly from the *RC* time constants of the devices, implying that additional improvements in the response speed can be achieved by greater optimization of the device architecture [45]. Additionally, it should be noted that as the modulation frequency applied to the irradiated laser light was increased, the observed rise and decay times began to decrease continuously in the frequency range investigated here, as shown in the inset of Figure 5b.

Finally, we investigated another important functionality of the PMMA passivation layer based on its excellent hydrophobicity, as shown in Figure 1b. Herein, the storage stability of the photodiodes was measured in terms of the photocurrent flowing through the photodiode devices. Given that the storage lifetime is closely linked to water permeation into the functional layers, particularly the MAPbI_3_ active layer, measuring the temporal responses of the stored photodiodes can provide information pertaining to the degradation mechanisms in these devices [50].

To measure the storage stability, the devices in this study were stored in a nitrogen glovebox between successive measurements of their temporal responses at a regular time interval of approximately 48 h. Temporal response measurements were also taken at zero bias at room temperature. Figure 5c presents the representative temporal response characteristics for the photodiodes stored for a total storage time of ~500 h. As shown in the figure, the photocurrent of the reference was reduced significantly to ~91.9% of its original value after being stored for approximately 500 h. In contrast, the sample exhibited a relatively negligible decrement (~0%) of the photo-response, even after the long storage time used here. Thus, as clearly shown in the figure, the sample with the cathode-side PMMA passivation layer is much more stable than the reference without a PMMA layer, indicating once again that the PMMA passivation layer significantly enhances the storage stability of MAPbI_3_ perovskite photodiodes. Additional details related to the long-term and/or continuous stability of the photodiode with the PMMA passivation layer, together with a more stable Ag, Cu, or Au cathode instead of the Al cathode used here, will be discussed elsewhere.

## 4. Discussion

The observations above highlight the effects of the cathode-side PMMA passivation layer; it not only improved the quality of the film but also decreased the leakage/noise currents through the MAPbI_3_ perovskite layers, thus providing a new paradigm of a high, fast, and durable performance of MAPbI_3_ perovskite-based photodiodes. Additional developments of organic/inorganic perovskite layers and/or the introduction of other novel functional layers will result in further improvements in the device performance for self-powered, highly sensitive, and stable solution-processable perovskite photodiodes with a cathode-side PMMA passivation layer.

## 5. Conclusions

In this study, we demonstrated that the introduction of a cathode-side PMMA passivation layer between an absorber layer of MAPbI_3_ and the ETL effectively suppressed the leakage/noise currents and in turn improved the performance of the self-powered solution-processable MAPbI_3_ photodiodes. With the cathode-side PMMA passivation layer, the interface quality of the MAPbI_3_ perovskite layer was shown to be improved, and the film defects were reduced. Moreover, we noted an increase in the grain size and effective decrements in the recombination losses as well as the noise currents. Consequently, even at zero bias voltage, the MAPbI_3_ photodiode with the PMMA passivation layer exhibited a notably high specific detectivity *D*^*^ of ~1.07 × 10^12^ Jones. Moreover, a significantly low *NEP* of ~230 fW/Hz^1/2^ and wide *LDR* of ~127 dB were achieved. The MAPbI_3_ photodiode with the PMMA passivation layer revealed excellent device performance compared to that of a conventional MAPbI_3_ photodetector without a passivation layer. A rapid rise response time of 50 μs and decay response times of 17 μs were also obtained, as were highly durable dynamic responses of the photodiode, even after a long storage time of nearly 500 h. These findings provide clear evidence of the remarkably improved device performance of a perovskite photodiode with a noise-reducing PMMA passivation layer at the interface between the MAPbI_3_ perovskite light-absorbing layer and the ETL. Thus, a cathode-side PMMA passivation layer in a perovskite photodiode, as presented here, can enhance the applicability of interface-engineered perovskites in photodiodes, imaging sensors, and various low-energy-consuming light-detecting devices.

## Figures and Tables

**Figure 1 nanomaterials-13-00619-f001:**
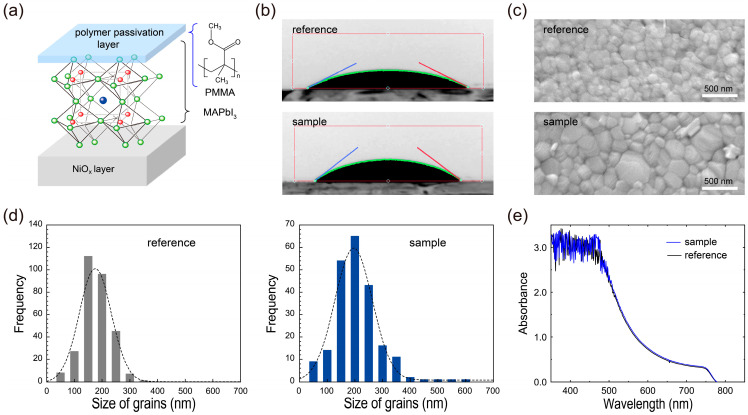
(**a**) Schematic illustration of the polymer passivation layer of PMMA on a MAPbI_3_ perovskite layer and corresponding chemical structure. (**b**) Comparisons of the water contact angles on the NiO_x_/MAPbI_3_ perovskite (reference) and NiO_x_/MAPbI_3_ perovskite/PMMA (sample) layers, corresponding to (**c**) high-magnification top-view SEM images and (**d**) grain size distributions derived from the SEM images. (**e**) UV–Visible optical absorption spectra of the reference and sample layers.

**Figure 2 nanomaterials-13-00619-f002:**
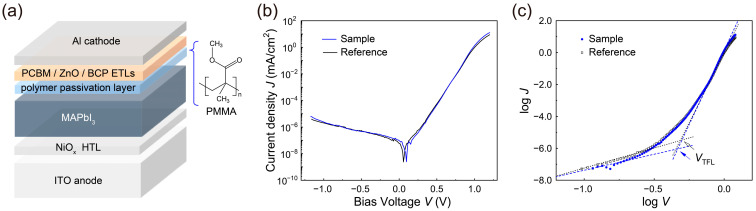
(**a**) Schematic device configuration of a MAPbI_3_ photodiode with a cathode-side PMMA passivation layer. Dark-current density-voltage (*J*_dark_-*V*) characteristics of MAPbI_3_ photodiodes without (reference) and with a cathode-side PMMA passivation layer (sample) on (**b**) a semi-log scale and (**c**) a log–log scale.

**Figure 3 nanomaterials-13-00619-f003:**
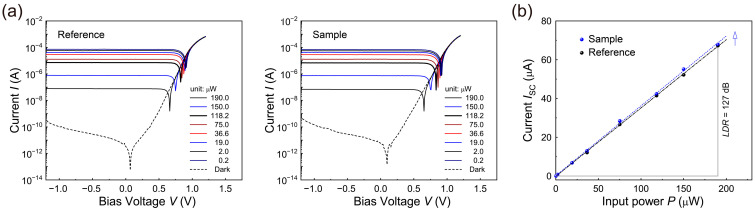
(**a**) Photocurrent characteristics of the MAPbI_3_ photodiodes of the reference (**left**) and sample (**right**) as a function of the bias voltage for several different intensity levels of irradiating light with a wavelength of 637 nm. (**b**) *I*_SC_ curves as a function of the input power of incident light (637 nm) for the reference and sample photodiodes.

**Figure 4 nanomaterials-13-00619-f004:**
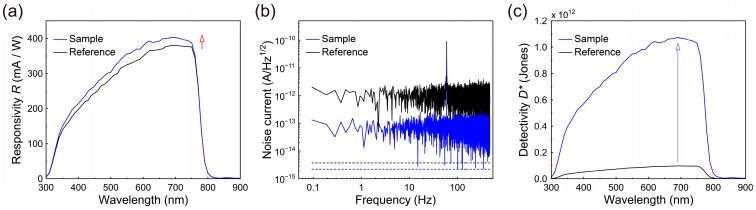
(**a**) Responsivity *R*_λ_ spectra at zero bias voltage, (**b**) noise currents vs. frequency characteristics, and (**c**) specific detectivity *D*^*^ spectra at zero bias voltage for the photodiodes of the reference and sample. The dotted lines in (**b**) show the shot noise levels for the reference (black) and sample (blue).

**Figure 5 nanomaterials-13-00619-f005:**
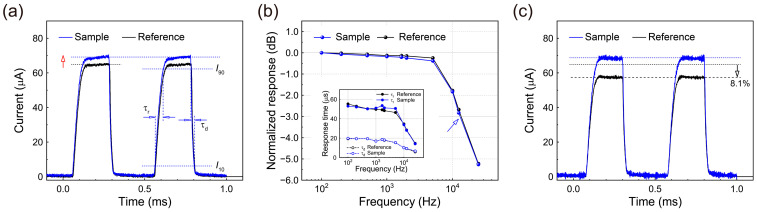
(**a**) Temporal photo responses of the reference and sample MAPbI_3_ photodiodes at zero bias voltage when turning the incident light (*λ* = 637 nm, *P* = 190 μW) on and off (2 kHz). The rise (*τ*_r_) and decay (*τ*_d_) times were determined as the time intervals between the 10% (*I*_10_) and 90% (*I*_90_) amplitude levels of the signal, respectively. (**b**) Normalized photocurrent versus the modulation frequency of incident light (*p* = 190 μW, 637 nm). The inset shows the dependences of *τ*_r_ and *τ*_d_ on the modulation frequency for the photodiodes studied here. (**c**) Temporal photo responses of the reference and sample photodiodes when stored for 500 h.

## Data Availability

Data presented in this study are available on request from the corresponding author.

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
