# Peer review of "Highly Efficient and Stable Self-Powered Perovskite Photodiode by Cathode-Side Interfacial Passivation with Poly(Methyl Methacrylate)"

_nanomaterials, 2023, doi:10.3390/nano13030619_

Round 1

Reviewer 1 Report

In the past few years, peroskite solar cells received more and more attention by the Scientifics. In this work, W. Kim et al. introduced a cathode-side passivation layer of poly(methyl methacrylate) (PMMA) for a highly efficient and stable self-powered CH3NH3PbI3 perovskite-based photodiode, which obtained some valuable results. The data support the conclusions. This work is suitable for this topic, which might be interesting for the readers. Based on my opinion, this work can be accepted after a minor-reversion.

1.     Figure 1d shows histograms of the grain size distributions of MAPbI3 perovskite layers without/with a PMMA passivation layer, but the author didn’t explain how to make it.

2.     The author investigates the optical absorption characteristics by UV-visible optical absorption spectra. However, the difference between the reference and sample layers is obscure.

3.     “To evaluating the photodiode performance over a wide light intensity range”, (page7, line 279), should be ‘To evaluate the photodiode performance over a wide light intensity range’.

4.     Page 2 , line 47-48: the author mentioned that “Due to these losses, much work remains for further improvements of the optoelectronic device performance and stability over conventional silicon-based devices”. When the author mentioned much work remains …, some references should be cited here such as Nano Energy 2022, 101, 107604 and Adv. Sustainable Syst. 2021, 2100244.

Reviewer 2 Report

Kim et al. demonstrated a nice photodiode by using PMMA passivated MAPbI3 as the active layers. The MAPbI3 photodiode with the PMMA passivation layer demonstrates a significantly high value of specific detectivity (~1.07 × 1012 Jones) and a wide linear dynamic response (~127 dB) with 20 rapid rise (~50 μs) and decay (~17 μs) response times. In addition, a higher stability is realized.

I would like to recommend its publication after some minor issues addressed:

1. The authors introduced PMMA dissolved in the antisolvent during spin-coating process. Did the authors optimize the concentration and dripping time? If so, could the authors comment the effect of the dripping conditions?

2. Since PCBM is coated with CB solvent, would this deposition partially dissolve PMMA coated on perovskite? This should be explained.

3. One thing interesting is Al cathode, due to its highly reactive ability, is it stable during continues measurement?

4. Some key references about passivation on perovskites are missed: doi: 10.1002/solr.202000244, DOI: 10.1039/C8EE01101J, DOI: 10.1039/D1EE00984B.

Reviewer 3 Report

This article reports about a strategy to cathode-side interfacial passivation with PMMA towards highly efficient and stable self-powered perovskite photodetector. A systematical comparison between the sample with PMMA passivation and the reference without PMMA passivation was carried out. The improvements in specific detectivity, responsivity, and noise current were obtained owing to the promoted MAPBI3 grain sizes and the suppressed perovskite-ETL interfacial defects. Overall, this manuscript can be accepted for publication after addressing the following concerns.

1)      The PMMA was introduced simultaneously as (but not after) coating the perovskite layer, so the discreted PMMA layer was not obtained. Is it right? If yes, the PMMA is not only present at the perovskite-ETL interface, but also inside of the perovskite layer. So the express of cathode-side interfacial passivation is not accurate.

2)      There is no direct evidence on the existence of PMMA layer. Cross-section and high-resolution SEM or TEM images with EDS mapping are suggested to supplied, and some other energy spectrum analyses on the NiOx/MAPbI3 perovskite and NiOx/MAPbI3 perovskite/PMMA layers are also should be implemented to confirm the PMMA existence.

3)      What are the thicknesses of NiOx HTL layer and PMMA layer?

4)      The preparation conditions of the devices, especially for the perovskite and PMMA, are optimized or not? And what is the optimization criterion?

5)      The temporal responses were investigated by irradiating monochromatic light modulated at a specific frequency (i.e., 2 kHz). How to modulate the chopped irradiation? The used frequency may have influence on the recorded rise and decay times. Besides, the determination of the rise and decay times should be indicated on the figures of temporal photoresponses.

6)      How to measure the 3 dB cutoff bandwidths? All the experiment methods on the shown experimental results should be reported clearly.   

Round 2

Reviewer 3 Report

Most of the comments have been well addressed, and the revised manuscipt can be acceptable for publication.